# Fabric Insert Injection Molding for the Preparation of Ultra-High Molecular Weight Polyethylene/High-Density Polyethylene Two-Component Self-Reinforced Composites

**DOI:** 10.3390/polym14204384

**Published:** 2022-10-17

**Authors:** Jian Wang, Da Wang, Qianchao Mao, Jinnan Chen

**Affiliations:** 1State Key Laboratory of Organic-Inorganic Composites, Beijing University of Chemical Technology, Beijing 100029, China; 2College of Mechanical and Electrical Engineering, Beijing University of Chemical Technology, Beijing 100029, China; 3School of Chemistry and Chemical Engineering, Beijing Institute of Technology, Beijing 100081, China

**Keywords:** injection molding, processing parameter, self-reinforced composite, polyethylene, mechanical properties

## Abstract

The fabric insert injection molding approach can be applied to produce easily recyclable self-reinforced polymer composites (SrCs) whose reinforcement and matrix are from the same polymer. However, the mechanical properties of the SrCs are usually limited due to the poor impregnation of the inserted fabric. In this work, the ultra-high molecular weight polyethylene (UHMWPE) fabrics were used as the insert, and the high-density polyethylene (HDPE) melt was injected to fill the mold cavity and impregnate the fabrics. The UHMWPE/HDPE two-component SrCs were prepared. The large difference of melting temperatures between UHMWPE and HDPE can establish a wide processing temperature window, and thus the impregnation of the fabric can be improved by increasing temperature. The tensile strength and modulus of the UHMWPE/HDPE SrCs were up to 148 and 1132 MPa, respectively. The peel strength could be up to 35.2 N/cm. The influences of four main injection molding parameters, including the injection temperature, injection pressure/packing pressure, injection velocity, and packing time, were investigated. The temperature, pressure, viscosity, and density of the matrix in the mold cavity were calculated by the numerical simulation to indicate the impregnation process during the fabric insert injection molding process.

## 1. Introduction

Injection molding is one of the most important manufacturing processes, especially for the mass production of identical plastic parts. It usually requires a metal mold with a cavity in the shape of the desired part, and the molten plastic is injected into the mold cavity for the forming and solidification [1]. The benefits of injection molding include short cycle time, low cost, various materials, compatible, diversely shaped parts, etc. Various sub-processes have added further capabilities to injection molding [2]. Insert injection molding, a subset of injection molding technologies, can produce various parts in automotive devices, medical devices, consumer goods, and cosmetics industry. It commonly uses metal insert, which is positioned inside the mold to produce plastic parts with metal attachment features. The insert injection molded part combines metal and plastic materials and can capitalize on the benefits of both. Moreover, insert injection molding can eliminate adhesives and assembling cost [3,4].

Nowadays, lightweight thermoplastic composites and their manufacturing processes have been developed to meet the requirements of aerospace, automotive, construction, etc. Various thermoplastic composites reinforced with short fibers, long fibers, glass mat, unidirectional fibers, fabrics, and nano reinforcements were applied [5]. Owing to the advantages of continuous fiber in high mechanical properties, unidirectional fibers and fabrics have been used a lot in automotive industry. Insert injection molding is becoming the common manufacturing method for the parts of continuous fiber-reinforced thermoplastic composites. The process can provide high specific stiffness and strength due to the continuous fibers and can also realize the production of parts with complex structures. Further advantages are the potential for a high level of function integration, net shape manufacturing, and large series production in the automotive industry [6]. Integrating injection molding with structural inserts of unidirectional (UD) fiber and woven fabric pre-impregnates has been gaining significant attention as a lightweight alternative to metal in various industrial fields [7,8]. However, two steps in manufacturing processes are needed for the woven fabric-reinforced thermoplastic composites [9]. Intermediate laminates or prepregs should be firstly prepared as the insert, and then the final products with the prepregs are formed through the insert injection molding. This two-step process increases the manufacturing cost and time. The fabrics can be directly inserted into the mold to save the prepregs preparation step [10], but the direct fabric insert injection molding generally cannot achieve quality standards required for consumer enclosures. In addition, the poor interfacial bonding between polymer and continuous fibers reduces the performance of the insert-molded composites [11].

If we change the conventional glass/carbon/natural fibers into polymer fibers and the fiber material is the same polymer as the matrix, the self-reinforced polymer composites (SrCs) will be formed. SrCs are also called single-polymer composites. They possess many advantages, such as lightweight, excellent interfacial bonding, high strength, and easy recyclability [12,13,14,15,16]. The main manufacturing process for SrC parts is compression molding [13,14,15,17,18,19,20,21], including hot compaction of fibers/tapes/fabrics, film stacking, coextrusion and compaction. The disadvantages of the compression molding for SrC parts include simple part geometry, deforming, reduced strength, long cycle time, high energy consumption, and cost. Some continuous manufacturing processes, such as extrusion-calendaring [22,23] and double belt pressing [24,25], have been also developed for the industrial production of SrC laminates or prepregs, which are generally intermediate materials. Whereas insert injection molding has been applied to produce SrC parts [26,27,28], complex shape and short cycle time can be realized. Good interfacial bonding between the inserted fabrics and the polymer melt can be realized due to the same polymer material. However, the matrix impregnation in the fibrous structural reinforcement is difficult because of the high viscosity of the thermoplastic polymer [29]. Moreover, the impregnation of the polymer fiber fabrics is much more difficult than that of the glass/carbon/natural fiber fabrics because the polymer fiber has a similar melting temperature with the polymer matrix and is easily molten at the processing temperature. Especially for injection molding, the injection temperature is usually 20–50 °C higher than the melting point of the polymer, and the higher injection temperature benefits the filling and impregnation of the fabrics but easily leads to fiber molten, and thus reduced strength. The fast cooling in the mold cavity is another factor resulting in poor impregnation and high porosity. Thereafter, the layer number and thickness of the fabric are limited in the fabric insert injection molding process for SrCs, which leads to low fiber volume fraction, and thus low mechanical strength. Therefore, the processing temperature window should be further enlarged and then a higher temperature could be applied to improve the impregnation. In addition, the impregnation mechanism of SrCs during the direct fabric insert injection molding is complicated and necessary to be investigated [30].

Two-component SrCs involve the same type of polymer with the same chemical composition but different chain configurations [15]. Examples include high-density polyethylene (HDPE)/low-density polyethylene (LDPE) SrCs, ultra-high molecular weight polyethylene (UHMWPE)/LDPE SrCs, UHMWPE/HDPE SrCs, and polypropylene (PP) homopolymer/PP copolymer SrCs. Due to the larger melting temperature difference between the two components, the two-component SrCs can have a wider processing temperature window than the one-component SrCs. UHMWPE as a subset type of polyethylene is composed of extremely long chains with a very high quality of parallel orientation and high level of crystallinity. UHMWPE woven fabric has the excellent features of high tensile strength, abrasion resistance, and cut resistance. It is capable of providing the newest generation of ballistic protection and safety materials. In this work, we applied the UHMWPE woven fabric as the reinforcement and HDPE as the matrix to prepare UHMWPE/HDPE two-component SrCs through insert injection molding. The UHMWPE fabrics were directly inserted into the mold cavity, and then the HDPE melt was injected to fill the cavity and penetrate the fabrics. Two layers of fabrics were used to increase the fiber volume fraction. The tensile properties of the SrCs prepared at different conditions were measured. The effects of different injection molding parameters, including injection temperature, velocity, pressure, and packing time, were discussed. The interfacial strength of the UHMWPE/HDPE SrCs prepared at the optimum condition was examined. Furthermore, in order to investigate the impregnation mechanism, numerical simulations of the fabric insert injection molding were conducted. The simulation results give strong support to the knowledge of the exact mechanism of the insert injection molding for SrCs.

## 2. Materials and Methods

### 2.1. Materials

HDPE granules (Marlex 9035, Chevron Philips Chemical Company LLC, Woodlands, TX, USA) with a density of 0.952 g/cm^3^ and a melt flow rate of 40 g/10min at 190 °C were used as the matrix. The melting point of the HDPE matrix was 126 °C, and the tensile strength at yield was 24 MPa. A commercial plain-woven UHMWPE fabric as the reinforcement was supplied by Barrday (Cambridge, ON, Canada). The fabric was woven from UHMWPE fibers (Spectra 900, Honeywell International Inc., Minneapolis, MN, United S). The areal density of the fabric was 231 g/m^2^, and the thickness was 0.43 mm. The fabric had 21 × 21 threads/inch in warp and weft directions. Each thread consisted of 300 individual fibers. The fiber density was 0.97 g/cm^3^, and the melting point of the fiber was 147 °C. The tensile strength and modulus of the fiber were 2.61 GPa and 79 GPa, respectively. Table 1 presents the main information of the materials used in the preparation of UHMWPE/HDPE two-component SrC samples.

### 2.2. Preparation Process

The fabric insert injection molding was conducted in an injection molding machine (SE-18D, Sumitomo Corporation, Tokyo, Japan). As shown in Figure 1, the experimental mold had a rectangular cavity with dimensions of 63.5 mm × 9.5 mm × 1.2 mm. The UHMWPE fabrics were first cut to 63.5 mm × 9.5 mm. Two layers of fabrics were, respectively, pasted on each side of the mold cavity walls. A fiber volume fraction of 40% could be obtained. The HDPE granulates were plasticized in the barrel at the injection temperature, and then the molten HDPE matrix was injected by the screw to fill the mold cavity and permeate the fabrics at the set injection velocity and injection pressure. After the packing and cooling stages, the SrC part with UHMWPE fabrics was finally ejected from the mold.

The effects of four injection molding parameters, including injection temperature, injection velocity, injection pressure, and packing time, were investigated. The set values of these four parameters are listed in Table 2. Several experiments were conducted to determine the set range of the parameters. The injection temperature ranged from 200 to 300 °C (Experimental No. 1–6). The maximum injection pressure of the injection molding machine was 30 kpsi. Since the cavity cannot be filled completely below 15 kpsi, 15~30 kpsi were chosen as the variation range of the injection pressure. The setting injection pressure was 15, 20, 25, and 30 kpsi, corresponding to 103.4, 137.9, 172.4, and 207 Mpa, respectively (Experimental No. 7–9, and 6). To ensure complete filling of the mold cavity, the injection temperature of 300 °C and the injection pressure of 30 kpsi were kept when investigating the effects of the injection velocity and the packing time. The injection velocity was changed from 3 to 11 in/s (0.076 to 0.28 m/s) (Experimental No. 10, 6, 11–13). The packing time was changed from 5 to 60 s (Experimental No. 12, 14–19). The packing pressure was the same as the set injection pressure. The other processing parameters were kept constant. The mold temperature was room temperature, around 25 °C. The filling time and cooling time were set to 1 s and 10 s, respectively.

### 2.3. Tensile Tests

The tensile properties of the UHMWPE/HDPE SrCs were measured by a universal testing machine (Instron 5166, Instron Corporation, Norwood, MA, USA). The rectangular SrC sample was cut to a dumbbell shape, according to the standard of DIN-53504. The tensile sample was fixed by the crosshead with a gauge length of 5 mm. The tensile test was conducted with a crosshead velocity of 5 mm/min at room temperature (around 25 °C). At least six samples were tested for each group.

### 2.4. Peel Test

The sample with sandwiched structure can be easily used in the T-peel test. A knife was used to cut open the end of the sample for about 8 mm and then the sample was loaded onto the universal testing machine. One opened part of the sample was clamped by the upper fixture, and the other part was clamped by the lower fixture. The crosshead velocity of 20 mm/min was applied. The average peel force was calculated from the data of each peak value of the measured load along the extension. The peel strength was calculated by dividing the average peel force by the unit width of the sample.

### 2.5. Numerical Simulation

Moldex 3D (CoreTech System Co., Ltd., Tainan, Taiwan) was applied to simulate the fabric insert injection molding process rheologically and thermally. The deformation of fabrics during the filling and packing stages was not considered, whereas the impregnation and the heat conduction were considered. A three-dimensional model was established, according to the samples prepared by the experiment, as shown in Figure 2. The model consists of sprue, runner, mold cavity, and two layers of UHMWPE fabrics. The geometric cavity was set as 63.5 mm × 9.5 mm × 1.2 mm. The thickness of the fabric was set as 0.4 mm considering the pressure effect. The fabric structure was set, according to the exact warp-weft structure of the UHMWPE fabric, and the fiber bundles were kept as an ensemble. The fabric volume content in the cavity was 40 vol%. The total mesh number was 2,368,680. Along the x, y, and z directions, 100 points were set as the measurement points to record the relative values of different parameters, including temperature, pressure, viscosity, etc.

The material data of a similar HDPE (Marlex 9018, Chevron Phillips Chemical Company LLC, Woodlands, TX, USA) were found in the database of the software. The material data for the UHMWPE fabrics (GUR 4113, Celanese Corporation, Irving, TX, USA) were also from the database. The governing equations for conservation of mass, momentum, and energy were applied, and the modified-Cross model with Arrhenius temperature dependence was used to describe the viscosity of the HDPE melt. These mathematical models can all be referred from the Moldex 3D Help. Process parameters of the injection molding were set, according to experimental conditions in Table 2.

## 3. Results and Discussion

### 3.1. Fabric Insert Injection Molding Process

The fabric insert injection molding process is shown in Figure 3a. The process conditions are: injection temperature of 300 °C, injection pressure of 30 kpsi, injection velocity of 9 in/s, and packing time of 15 s. The cavity was completely filled at 0.04 s. Unlike the fountain flow pattern in the conventional injection molding process, the matrix flow was separated into two streams due to the structure with two upper and lower layers of fabrics. The fabrics were impregnated during the injection process. Figure 3b shows the temperature distribution on the middle sections of the cavity in the x and y directions, respectively. At the end of the filling time (0.04 s), the highest temperature is mainly located in the middle, and it became 304 °C due to the thin space shearing, whereas the temperature at the interface of the fabrics was still very low, around 110 °C. The temperature around the surface of the fabrics was about 150 °C, which is higher than the melting temperature of the UHMWPE fiber. Thus, the fabrics were partially molten, which benefits the interfacial adhesion between the matrix and the fabrics. At the end of the packing time (16 s), the temperature decreased a lot, and the interfacial temperature was maintained at around 100 °C. This indicates that the matrix and the fabrics were bonded together. At the end of the cooling time (27 s), the residual temperature was mainly concentrated in the center of the mold cavity (60 °C) and the center of each fiber bundle (65 °C). The residual temperature at the center of the fiber bundles was higher than in the other positions. This indicates that the heat transferred to the center of the fiber bundles during the packing stage dissipated more slowly in the reinforcement.

Figure 4 shows the variations of temperature, viscosity, and pressure with time along the x direction. In Figure 4a, the temperature decreased. The viscosity increased rapidly and became stable at about 4 s. The temperature was a little higher at x4 near the gate during the filling stage because the gate was close to the melt resource of the injection unit with the constant injection temperature. In Figure 4b, higher pressure can be maintained at the position close to the gate, in comparison to the other positions far away from the gate. Figure 5 shows the variations of temperature, viscosity, and pressure with time along the y direction. During the injection and packing stages, the temperature in the middle of the cavity (y2–y4) was higher, and the relative viscosity was lower. The pressure in the middle (y3) and close to the cavity wall (y1 and y5) was higher than that in the two streams (y2 and y4), which corresponds to the two streams phenomena in Figure 3a. Figure 6 shows the variations of temperature, viscosity, and pressure with time along the z direction. The temperature in the fabrics decreased extremely fast. The temperature at the interface between the fabric and the matrix was higher and can be kept at around 80 °C during the packing stage for a while. This indicates that the fabrics would not be easily molten due to the very short time. The temperature in the middle decreased relatively slowly. The pressure at z1 and z5 decreased rapidly. It indicates that the impregnation of fabrics was only subjected to high pressure at the moment of the filling stage.

### 3.2. Effects of Injection Molding Parameters

When the injection temperature was lower than 200 °C, the HDPE melt did not completely fill the cavity. When the temperature was higher than 300 °C, the melt became yellow and degradation occurred. Therefore, the injection temperature range was between 200 and 300 °C. Figure 7a shows the tensile strength and modulus of the UHMWPE/HDPE SrCs prepared at different injection temperatures. The injection temperature of 200 °C is too low to complete the impregnation of the fabrics even though the cavity was filled. At a lower injection temperature, the viscosity of the HDPE melt is higher. The polymer melt cools quickly during the filling stage, which hinders the penetration of the melt into the fabric, and thus leads to a poor bonding between the fiber and the matrix. With the increase in injection temperature, the tensile strength and modulus of the UHMWPE/HDPE SrCs increased, which is because the decrease in melt viscosity enhanced the permeability of the melt into the fabrics. It is noted that the average tensile strength remained constant from 220 to 260 °C. This indicates that the improvement of impregnation was little by the increase in melt temperature in the range from 220 to 260 °C. From 260 to 300 °C, a significant increase in tensile strength indicates the improvement of interface bonding and impregnation. When the injection temperature was raised to 300 °C, the tensile strength and modulus of the UHMWPE/HDPE SrCs reached the maximum values of 110 ± 4 and 720 ± 15 MPa, respectively. This is the reinforcing effect brought by the high-strength UHMWPE fiber and the good interfacial adhesion between the fibers and matrix. Although the injection temperature of 300 °C is very high, rapid cooling leads to only a partial melting of the fabrics. Figure 3b shows that the interfacial temperature between the fabrics and the matrix could be around 150 °C. Therefore, the fabric insert injection molding method could obtain a wide processing temperature window. However, the degradation of HDPE has limited the increase in injection temperature. Otherwise, the improvement of impregnation through further increasing injection temperature may lead to a higher tensile strength.

Figure 7b shows the total temperature of the matrix and the matrix with insert at the end time of the filling stage. The temperature decreases after filling, and the residual temperature is higher at higher injection temperatures. However, the result shows that the total temperature of the matrix with insert ranged between 30 to 120 °C and was influenced little by the injection temperature. The initial temperature of the UHMWPE fabrics was the mold temperature (25 °C). During the filling stage, the HDPE melt with high temperature filled into the mold cavity and penetrated the fabrics, while the UHMWPE fabric insert was heated up due to the heat transfer in the mold cavity. The variation of the total temperature of the matrix and insert in the cavity is mainly dependent on the heat transfer between the matrix and the insert and the heat conduction of the metal mold. The total temperature of the matrix and the insert, shown in Figure 7b, indicates that the heat transferred from the matrix to the insert was not enough to change the temperature of the whole cavity, even though the injection temperature increased in a large range from 200 to 300 °C.

Figure 7c shows the viscosity was affected significantly by the injection temperature. The variation of injection temperature from 200 to 300 °C led to a decrease of 1889 Pa·s of viscosity. The sample density at the end time of the packing stage in Figure 7d varied between 0.905 and 0.907 g/cm^3^. In comparison with the exact densities of the HDPE matrix and the UHMWPE fabric, the sample density at the end time of the packing stage was lower because the solidification was not completed. The sample density is mainly dependent on the packing process. The density difference of 0.002 g/cm^3^ in Figure 7d indicates little influence of the injection temperature on the density.

The woven structure of the fabric is another important factor affecting the permeability. A tight and thick woven structure is usually difficult to be impregnated by the thermoplastic melt. In this work, the thickness of the fabric was 0.43 mm. According to the thickness of the mold cavity, 1.2 mm, the two layers of the fabrics have taken 1/3 of the cavity. The fiber volume fraction can be kept high, and thus a good composite strength can be realized. However, it increased the difficulty of filling and impregnation. Therefore, high injection and packing pressure are necessary. The experiments showed that the melt failed to fill the cavity completely when the injection pressure was less than 103.5 MPa. According to the upper limit of the pressure value that can be reached by the injection molding machine, the injection pressure range between 103.5 to 207 MPa was selected for experiments. However, due to the thin cavity left for the filling of polymer melt and the fast cooling during the filling stage, the actual injection pressure was lower than the set values. The actual pressure of 62.1, 86.3, 107.6, and 120.8 MPa corresponded to the set injection pressure of 103.4, 137.9, 172.4, and 207 MPa, respectively.

Figure 8a shows the tensile strength and modulus of the UHMWPE/HDPE SrCs prepared at different injection pressure. The actual injection pressure, rather than the set injection pressure, was used in Figure 8a. The tensile strength and modulus increased with increasing injection pressure because higher injection pressure is conducive to improving the impregnation. However, a higher pressure leads to a tighter and thinner fabric that will resist impregnation. Higher temperature and lower viscosity can cooperate with the higher pressure, and thus the bonding adhesion between the fibers and matrix can be improved. Figure 8b,c show that the temperature in the cavity was low and the viscosity was high at the lowest injection pressure. However, the other higher injection pressure influenced little on the temperature and viscosity. Figure 8d also shows the little influence of the injection pressure on the density of the matrix. The increase in injection pressure can increase the impregnation, and thus the tensile strength and modulus, but too high an injection pressure will not improve these significantly.

The mold temperature is at room temperature, thus, the thermoplastic melt cools faster when it fills the cavity. The viscosity will increase and the impregnation process may fail. Therefore, fast injection is needed in the fabric insert injection molding. The injection velocities ranging from 0.076 to 0.28 m/s were selected in this work. Figure 9a shows the tensile properties of the UHMWPE/HDPE SrCs changed with different injection velocities. The tensile strength and modulus of the UHMWPE/HDPE SrCs increased and then decreased. The highest strength and modulus were 117 ± 3 and 1005 ± 25 MPa, respectively. The fast injection filling process gives more time for keeping the high melt temperature and low viscosity, which benefits the impregnation in the packing stage. As with the discussion in Section 3.1, the shear rate is higher at a higher injection velocity, which results in the decrease in viscosity. In particular, the thin cavity gives the opportunity for a high shear rate. With the combination of high temperature and shear rate, the viscosity can be decreased. Thereby, the permeability is enhanced, and the tensile strength and modulus were increased. However, excessive shear will generate friction heat and increase the temperature, then fiber molten and degradation will occur. That is the reason why the tensile strength and modulus decreased when the injection velocity increased to 0.28 m/s. Another reason for the decrease in the tensile strength and modulus is probably the interface damage due to the displacement of the partially molten fibers at the much higher injection velocity. Figure 9b,c show that the temperature of the matrix and the matrix with insert increased and the viscosity decreased with increasing injection velocity. The viscosity changed little and even increased a little when the injection velocity increased from 0.23 to 0.28 m/s. This can also explain the decrease in tensile strength and modulus. The density of the matrix after packing in Figure 9d changed little at different injection velocities.

Figure 10a shows the change of tensile strength and modulus of the UHMWPE/HDPE SrCs with packing time. The tensile strength and modulus increased with the packing time increasing from 5 to 15 s and then remain unchanged. At the packing time of 15 s, the tensile strength and modulus of the composites reached the maximum values. Due to the cooling and shrinkage of the material in the cavity, the screw continues to move forward slowly in the packing stage after the completion of mold filling. The amount of polymer melt increases with the extension of packing time. However, no more polymer melt enters the cavity when the gate freezes. The crystallization rate of PE is fast, and PE can be cooled and crystallized in a short time. It is clearly seen that the HDPE at the gate solidified at the packing time of 15 s. The mass of materials in the cavity would not change, and thus the tensile properties of the composites did not change. Figure 10b indicates that the temperature of the matrix after the filling stage decreased little with increasing packing time. The total temperature of the matrix with insert decreased only at the much long packing time of 60 s. The viscosity in Figure 10c increased a little from 72.3 to 86.9 Pa·s, but it was not influenced by the packing time because the viscosity values were from the data at the end time of filling stage. The significant increase in the density of the matrix in Figure 10d indicates the increasing mass of matrix during the packing stage, corresponding to the packing time from 5 to 20 s.

### 3.3. Optimum Properteis of the UHMWPE/HDPE SrCs

At the optimum condition of injection temperature of 300 °C, injection pressure of 30 kpsi, injection velocity 9 in/s, and packing time of 15 s, the tensile strength and modulus of the UHMWPE/HDPE SrCs were up to 148 ± 3 and 1132 ± 17 MPa, respectively. Figure 11 shows the tensile stress–strain curves of the UHMWPE/HDPE SrCs, in comparison with the unreinforced HDPE. The yield stresses of the UHMWPE/HDPE SrCs and unreinforced HDPE were 151 and 20 MPa, respectively. The yield stress of the UHMWPE/HDPE SrCs is 7.6 times that of the unreinforced HDPE. The elongation at break of the UHMWPE/HDPE SrCs was 0.33, which is much lower than that of the unreinforced HDPE. The HDPE resin is a tough material. There was a large plastic deformation area in the tensile stress–strain curve of the unreinforced HDPE. The stress decreased after yield and then remained unchanged with the increase in strain, so the stress curve had a plateau area. The tensile stress response to the strain of the UHMWPE/HDPE SrCs was different. There was no plateau area in the stress curve. The stress increased linearly but slowly with the increase in strain and then increased fast when the strain increased over 0.05. This is due to the structure of the woven fabric. At the beginning of the tensile test of the UHMWPE/HDPE SrCs, the matrix with the fabrics was firstly stretched. The fabric structure took time to be stretched straightly with the matrix. When the fiber bundles were fully extended, the fibers began to bear the tensile load. Thus, the tensile modulus was much higher than that of the unreinforced HDPE. After the yield point, a region of plastic deformation appeared before breaking. The plastic deformation indicates the high toughness of PE.

Figure 12 shows the peel load curve of the UHMWPE/HDPE SrCs as a function of extension. The peeling force fluctuated because of the uneven surface structure of the fabric. With the increase in extension, the peeling force increased. During the injection molding process, the HDPE melt was injected from the gate to the end of the mold cavity. Under the effect of heat conduction on the cold wall surface of the cavity, the melt cooled rapidly. The temperature at the gate was always higher than that at the end of the cavity. The decreasing temperature along the filling pathway of the melt influenced the interfacial adhesion between the fabric and the matrix. Thereafter the interfacial strength of the sample close to the gate was higher than that close to the end of the cavity. The average interfacial strength of the UHMWPE/HDPE SrCs was 35.2 N/cm.

The comparison information of optimal tensile properties of UHMWPE reinforced PE SrCs produced by different manufacturing methods is listed in Table 3. The mechanical properties of SrCs are generally affected by basic material strength, reinforcement structure, fiber fraction, and processing condition. In comparison with HDPE, the LDPE and ultra-low molecular weight PE (ULMWPE) as the matrix [23,31,32,33] lead to lower mechanical properties. In comparison with short fiber and fabric structures, the unidirectional fiber reinforced structure leads to higher mechanical properties along the fiber stretching direction [34,35,36], whereas fabric structure can undertake two directions of load. A higher fiber volume fraction can improve mechanical properties [36]. The compounding and injection molding method of UHMWPE/HDPE [37,38,39,40] is another potential process to be an industrial level. However, the preparation of the UHMWPE/HDPE/PE reactor blend should be prepared firstly, and the additive synthetic and catalyst are needed [38,39,40]. Therefore, the cost and the industrial route for the compounding and injection molding method need further discussion. For the fabric insert injection molding method in this work, the plain fabric structure can confirm a good reinforcement type, and then two layers of fabrics can confirm a high fiber volume fraction. The comparison results, including the tensile strength, modulus, and break strain, present a significant improvement through the fabric insert injection molding.

## 4. Conclusions

The fabric insert injection molding process was conducted to prepare UHMWPE/HDPE SrCs. The two-component concept gives a wide processing temperature window, so that a higher temperature could be used to further improve the impregnation of the fabrics. Moreover, the fiber volume faction can be increased, and thus the mechanical properties of the SrCs could be enhanced. Two layers of UHMWPE fabrics were used, and thus a fiber volume fraction of 40 vol% was realized. The effects of process parameters, such as injection temperature, injection/packing pressure, injection velocity, and packing time, on the tensile properties of UHMWPE/HDPE SrCs and the impregnation of fabrics were studied by experiments and numerical simulation. The results showed the optimum processing condition: injection temperature of 300 °C, injection pressure of 30 kpsi, injection velocity 9 in/s, and packing time of 15 s. The tensile strength of the UHMWPE/HDPE SrCs reached the maximum value of 151 Mpa, which is 7.6 times that of unreinforced HDPE. The average peel strength of 35.2 N/cm indicates good interfacial adhesion between the fabrics and matrix. The increase in injection temperature, injection/packing pressure, injection velocity, and packing time can all improve the interfacial bonding between the fabrics and matrix, and thus can improve tensile properties. However, the injection temperature should not be higher than the degradation temperature of the matrix. The increase in injection/packing pressure is dependent on the ability of the injection molding machine. The injection velocity should be set considering the displacement of the inserted fabrics. The optimum packing time is dependent on the frozen time of the sprue. The optimum tensile property of the UHMWPE/HDPE SrCs is comparable with those prepared by the other methods. The numerical simulation results gave much support to knowing the mechanism of the fabric insert injection molding process. The numerically calculated data were successfully used to explore the changes of various parameters, such as temperature, pressure, viscosity, and density in the mold cavity, and then clarify the influence of different process parameters.

## Figures and Tables

**Figure 1 polymers-14-04384-f001:**
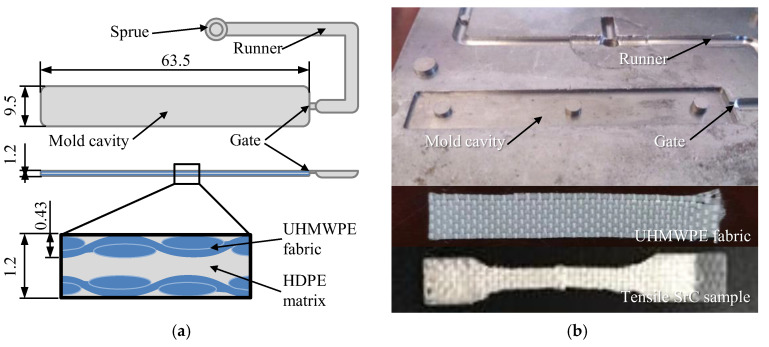
Schematics of the mold cavity and the fabric insert injection molding structure of the UHMWPE/HDPE SrC sample (**a**) and pictures of the mold plate, fabric, and the tensile SrC sample (**b**) [28].

**Figure 2 polymers-14-04384-f002:**
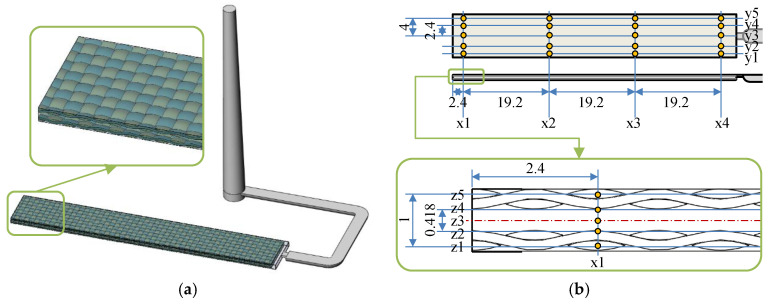
Simulation model of the fabric insert injection molded part with sprue and runner (**a**) and the positions of measurement points in the model (**b**).

**Figure 3 polymers-14-04384-f003:**
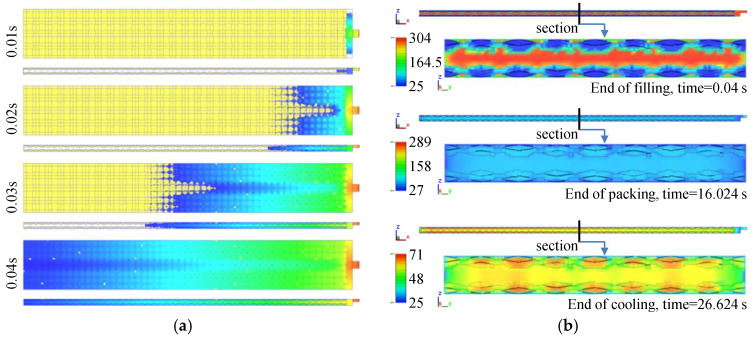
Numerically simulated fabric insert injection molding process (**a**) and the temperature distribution on the middle sections of the cavity in the x and y directions at the end of filling, packing, and cooling (**b**).

**Figure 4 polymers-14-04384-f004:**
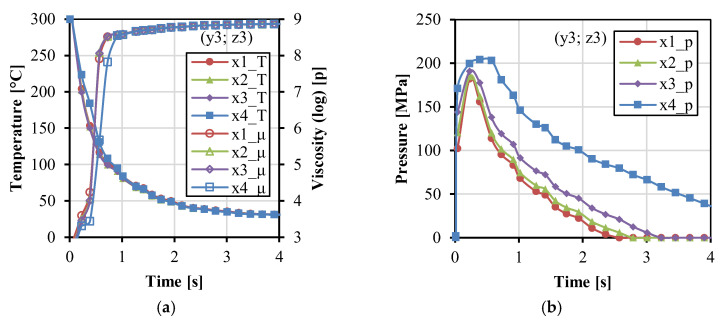
Parameter values as a function of time at the four points (y3 = 0, z3 = 0.6) along the x direction: temperature, viscosity (**a**), and pressure (**b**).

**Figure 5 polymers-14-04384-f005:**
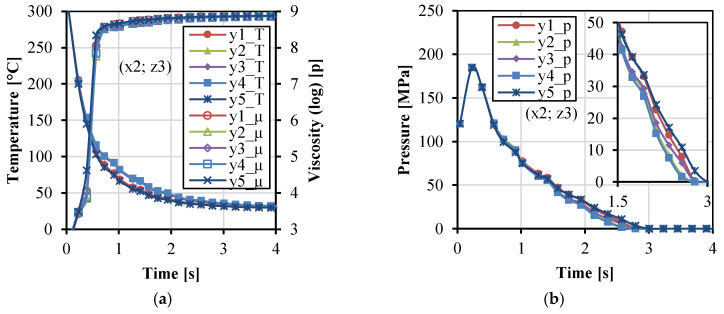
Parameter values as a function of time at the five points (x2 = 21.6, z3 = 0.6) along the y direction: temperature, viscosity (**a**), and pressure (**b**).

**Figure 6 polymers-14-04384-f006:**
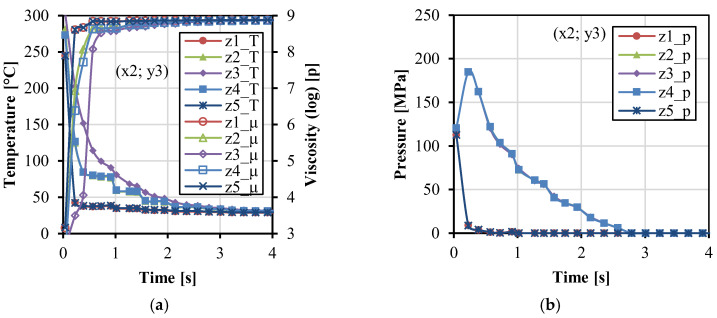
Parameter values as a function of time at the five points (x2 = 21.6, y3 = 0) along the z direction: temperature, viscosity (**a**), and pressure (**b**).

**Figure 7 polymers-14-04384-f007:**
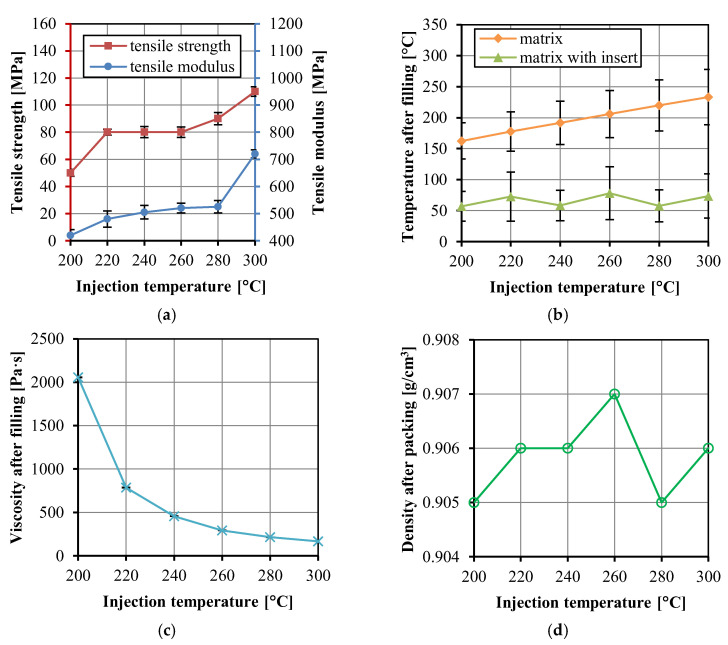
Tensile strength and modulus of the UHMWPE/HDPE SrCs prepared at different injection temperatures (**a**), the numerically calculated temperature in the cavity after the filling stage (**b**), the numerically calculated viscosity of the matrix in the cavity after the filling stage (**c**), and the density of the matrix after the packing stage (**d**).

**Figure 8 polymers-14-04384-f008:**
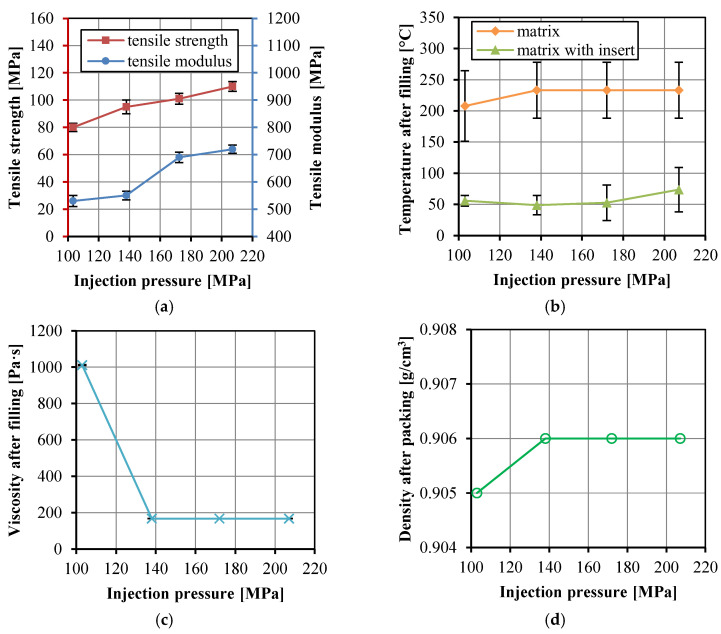
Tensile strength and modulus of the UHMWPE/HDPE SrCs prepared at different injection pressure (**a**), the numerically calculated temperature in the cavity after the filling stage (**b**), the numerically calculated viscosity of the matrix in the cavity after the filling stage (**c**), and the density of the matrix after the packing stage (**d**).

**Figure 9 polymers-14-04384-f009:**
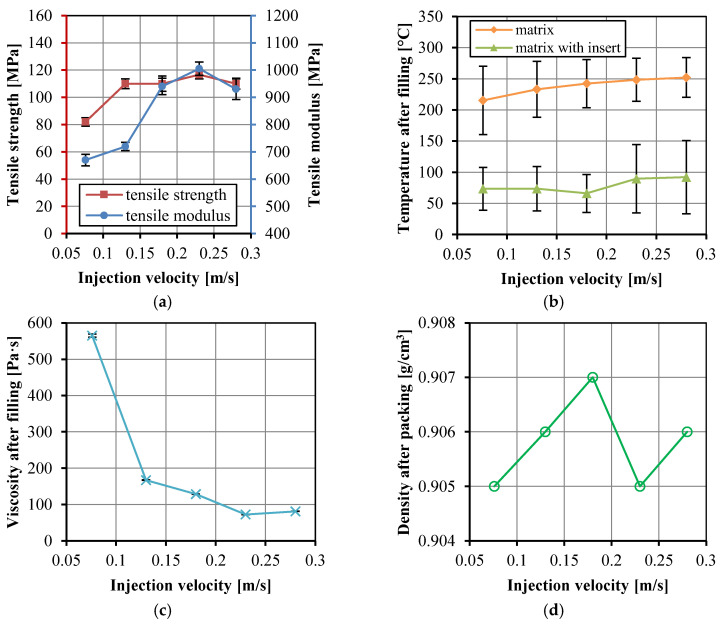
Tensile strength and modulus of the UHMWPE/HDPE SrCs prepared at different injection velocities (**a**), the numerically calculated temperature in the cavity after the filling stage (**b**), the numerically calculated viscosity of the matrix in the cavity after the filling stage (**c**), and the density of the matrix after the packing stage (**d**).

**Figure 10 polymers-14-04384-f010:**
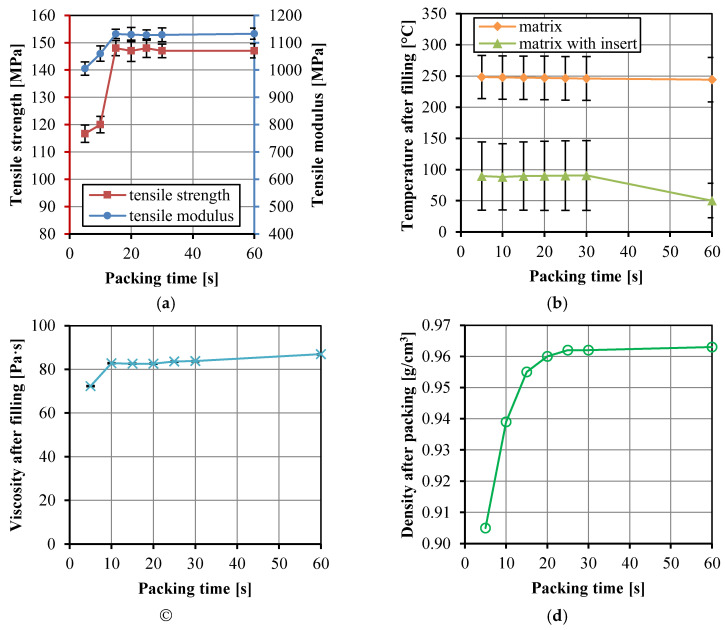
Tensile strength and modulus of the UHMWPE/HDPE SrCs prepared at different packing time (**a**), the numerically calculated temperature in the cavity after the filling stage (**b**), the numerically calculated viscosity of the matrix in the cavity after the filling stage (**c**), and the density of the matrix after the packing stage (**d**).

**Figure 11 polymers-14-04384-f011:**
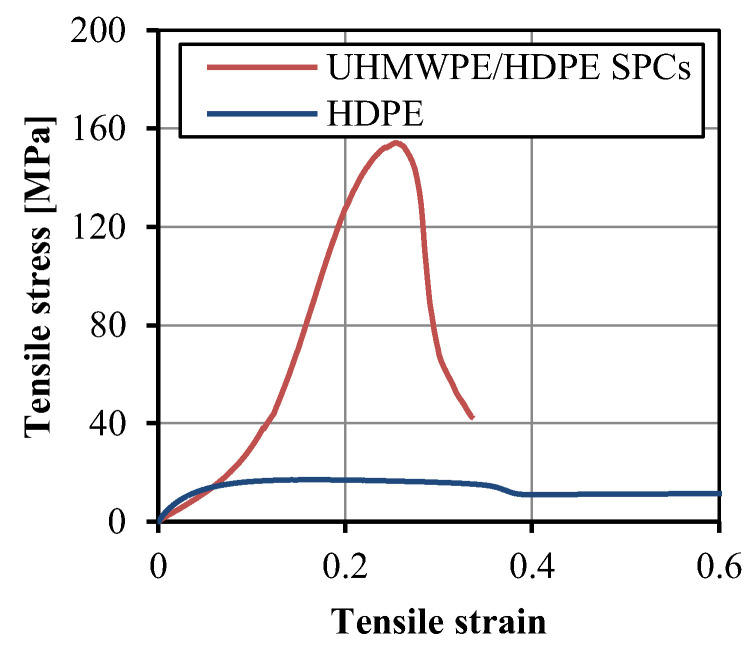
Tensile stress–strain curves of the tensile samples of the UHMWPE/HDPE SrCs and unreinforced HDPE.

**Figure 12 polymers-14-04384-f012:**
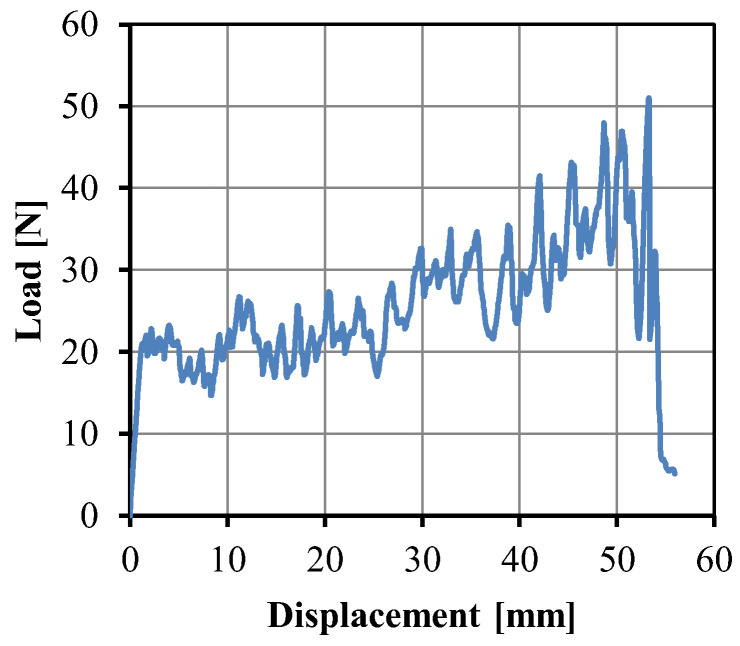
Peel load curve of the UHMWPE/HDPE SrCs as a function of displacement.

**Table 1 polymers-14-04384-t001:** Material information for the preparation of UHMWPE/HDPE two-component SrC samples.

Component	Material	Brand	Melting Point (°C)	Density (g/cm^3^)	Tensile Strength (MPa)
Matrix	HDPE	Marlex 9035	126	0.952	24
Reinforcement	UHMWPE fiber	Spectra 900	147	0.97	2610, single fiber 300, fabric

**Table 2 polymers-14-04384-t002:** Injection molding parameters for the preparation of the UHMWPE/HDPE SrC samples.

Experimental No.	Injection Temperature (°C)	Injection Pressure (kpsi)	Injection Velocity (in/s)	Packing Time (s)
1	200	30	5	5
2	220	30	5	5
3	240	30	5	5
4	260	30	5	5
5	280	30	5	5
6	300	30	5	5
7	300	15	5	5
8	300	20	5	5
9	300	25	5	5
10	300	30	3	5
11	300	30	7	5
12	300	30	9	5
13	300	30	11	5
14	300	30	9	10
15	300	30	9	15
16	300	30	9	20
17	300	30	9	25
18	300	30	9	30
19	300	30	9	60

**Table 3 polymers-14-04384-t003:** Comparison of optimal tensile properties of UHMWPE-reinforced PE SrCs produced by different methods.

Reference	Materials	Methods	Type of Reinforcement	Fiber Fraction	Tensile Strength [MPa]	Tensile Modulus [MPa]	Break Strain [%]
This work	UHMWPE/HDPE	Insert injection molding	Plain fabric	20 wt%	148	1132	33
[31]	UHMWPE/LDPE	Insert injection molding	Plain fabric	5 wt%	23.8	-	60
[32]	UHMWPE/LDPE	Compression molding (Film stacking)	Plain fabric	14 vol%	70.1	2354	<5
[23]	UHMWPE/LDPE	Extrusion-calendering	Plain fabric	11 vol%	78.8	676.6	-
[33]	UHMWPE/ULMWPE	Compounding and injection molding	Tailored blend	90.2 wt%	65.5	1248.7	83.9
[34]	UHMWPE	Compression molding (Powder impregnation)	Cross-ply laminates	5 wt%	56	1400	-
[35]	UHMWPE	Compression molding (Powder impregnation)	Short fiber	30 wt%	65.2	2260	16
[36]	UHMWPE	Compression molding (Hot compaction)	Unidirectional fibers	~100%	460 ± 12	21.1 ± 0.8	<5
[37]	UHMWPE/HDPE	Compounding and injection molding	Chopped and irradiated fibers	20 wt%	41.1	1620	17.1
[38]	UHMWPE/HDPE wax/PE reactor blend	Compounding and injection molding	Tailored blend	12 wt%	134.2	3770	12
[39]	UHMWPE/HDPE/PE reactor blend	Compounding and injection molding	Tailored blend	24 wt%	160	4200	<5
[40]	UHMWPE/HDPE reactor blends prepared with CrQCp and nBuZr	Compounding and injection molding	Tailored blend	18.6 wt%	171	2345	13.1

## Data Availability

The datasets generated and analyzed during the current study are available from the corresponding author on reasonable request.

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
