# Peer review of "Fabric Insert Injection Molding for the Preparation of Ultra-High Molecular Weight Polyethylene/High-Density Polyethylene Two-Component Self-Reinforced Composites"

_polymers, 2022, doi:10.3390/polym14204384_

Round 1

Reviewer 1 Report

The manuscript entitled “Fabric Insert Injection Molding for the Preparation of Ultra High Molecular Weight Polyethylene/High-Density Polyethylene Two-Component Self-reinforced Composites” described the preparation of Self-reinforced Composites by using Ultra High Molecular Weight Polyethylene as a reinforcement insert and High-Density Polyethylene melt that is being injected into the mold. However, the manuscript suffers from a number of shortcomings as described below: 

1.        The authors have listed the injection molding parameters in Table 2. It is mentioned that the injection temperature varies from 200 to 280°C (Experiment 1 to 6) but the injection temperature for experiment 6 is 300°C. Please explain. (Page 4)

2.       Is there any particular reason behind fixing the injection pressure, injection speed and packing time as 30kpsi, 5in/s, 5s respectively for Experiment 1-6. Reason should be included for better clarity.  (Page 4 -Table 2)

3.       The authors have mentioned that the cavity cannot be filled when the injection pressure is less than 15kpsi.Then why is the variation range of injection pressure given as 5 to 30kpsiand there is no supporting data (Page 4)

4.       It is necessary that the authors mention that holding time is also known as packing time for clear understanding. (Page 4)

5.       Any particular reason that the packing pressure is given as same as that of the injection pressure? (Page 4)

6.       The authors have mentioned that the other parameters have been kept constant. Enlist the parameters. (Page 4)

7.       Section 2.4: Peel test -Methodology not clear -The first four lines have to be re-written.

(Page 5)

8.        Since Figure 4 has both a and b images explanation should be given separately for both.

 (Page 7)

9.       The authors have mentioned that in Figure 4 a that the temperature was a little higher at x4 near the gate. Why and how is it deciphered?

1The authors have mentioned that the tensile strength and tensile modulus increases with respect to temperature. But in Figure 7a the tensile strength remains constant from 220 to 260°C. (Page 9)

1The discussion of the Figures 7b (matrix with insert) and 7d (change in density with respect to temperature) should be more clearly written for better interpretation

TThe authors have mentioned that the tensile strength and modulus of the self-reinforced composites increases and then decreases. The reason for this should also be cited.

(Figure 9a) (Page 11)

1The authors have mentioned that the viscosity increases with time in Figure 10c. But the graph does not show very much of an increase. It should be clearly discussed. (Page 12)

Author Response

Thank you very much for the review comments. Accordingly, we have revised the manuscript to address the comments from the reviewers and also improved the writing quality. The detailed point-by-point responses are attached as follows:

  1. The authors have listed the injection molding parameters in Table 2. It is mentioned that the injection temperature varies from 200 to 280°C (Experiment 1 to 6) but the injection temperature for experiment 6 is 300°C. Please explain. (Page 4)

Response 1: we are sorry for the mistake, it should be “200 to 300 °C”.

  1. Is there any particular reason behind fixing the injection pressure, injection speed and packing time as 30kpsi, 5in/s, 5s respectively for Experiment 1-6. Reason should be included for better clarity. (Page 4 -Table 2)

Response 2: Experiments 1-6 were mainly applied to know the effect of injection temperature, therefore the other three parameters were fixed. In order to ensure complete filling of the mold cavity, several experiments were conducted. Because the low temperature does not benefit the filling, so the lowest temperature of 200 °C was used and the other three parameters were changed. It was found that the injection pressure of 30 kpsi, the injection speed of 5 in/s, and the packing time of 5 s can realize the complete cavity filling, although these values were not optimum. Thus, the following experiments were conducted to confirm the other parameters. Additionally, it was found that the high injection pressure of 30 kpsi (the maximum pressure of the injection molding machine) was important for the complete filling of the mold cavity.

  1. The authors have mentioned that the cavity cannot be filled when the injection pressure is less than 15kpsi.Then why is the variation range of injection pressure given as 5 to 30kpsiand there is no supporting data (Page 4)

Response 3: We are sorry for the mistake, it should be “15 to 30 kpsi”.

  1. It is necessary that the authors mention that holding time is also known as packing time for clear understanding. (Page 4)

Response 4: Thanks. We have changed the “holding time” in this manuscript into “packing time”.

  1. Any particular reason that the packing pressure is given as same as that of the injection pressure? (Page 4)

Response 5: It was found that the actual injection pressure cannot be up to the set injection pressure due to the very short injection time, but the mold filling needs high pressure, so the set packing pressure was kept the same as the set injection pressure.

  1. The authors have mentioned that the other parameters have been kept constant. Enlist the parameters. (Page 4)

Response 6: The other main parameters are the mold temperature, filling time, and cooling time, they were given at the end of the paragraph on page 4. “The mold temperature was room temperature around 25 °C. The filling time and cooling time were set to 1 s and 10 s, respectively.”

  1. Section 2.4: Peel test -Methodology not clear -The first four lines have to be re-written. (Page 5)

Response 7: thanks for the suggestion. We have modified the section of the peel test and clarified the test method, especially the preparation of the sample with an opened end.

  1. Since Figure 4 has both a and b images explanation should be given separately for both. (Page 7)

Response 8: the text for explanation has been separated. Please see the text in red color.

  1. The authors have mentioned that in Figure 4 a that the temperature was a little higher at x4 near the gate. Why and how is it deciphered?

Response 9: The temperature was a little higher at x4 near the gate during the filling stage because it was close to the melt resource of the injection unit which was at the constant injection temperature.

  1. The authors have mentioned that the tensile strength and tensile modulus increases with respect to temperature. But in Figure 7a the tensile strength remains constant from 220 to 260°C. (Page 9)

Response 10: Thanks for this question. The average tensile strength remained constant from 220 to 260 °C, indicating the improvement of impregnation was little by the increase of melt temperature in the range from 220 to 260 °C. From 260 to 300 °C, the significant increase in tensile strength indicates the improvement of interface bonding and impregnation.

  1. The discussion of the Figures 7b (matrix with insert) and 7d (change in density with respect to temperature) should be more clearly written for better interpretation

Response 11: We have modified the text of the discussion on Figures 7 (b) and (d). Please see the text in red color.

  1. The authors have mentioned that the tensile strength and modulus of the self-reinforced composites increases and then decreases. The reason for this should also be cited. (Figure 9a) (Page 11)

Response 12: We have modified the text of the discussion. Please see the text in red color.

  1. The authors have mentioned that the viscosity increases with time in Figure 10c. But the graph does not show very much of an increase. It should be clearly discussed. (Page 12)

Response 13: Thanks for this comment. The increase of viscosity in Figure 10 (c) is from 72.3 to 86.9 Pa·s but it was not influenced by the packing time, because the values were from the data at the end time of the filling stage. So we modified the text of the discussion. Please see the text in red color.

Reviewer 2 Report

The present manuscript entitled “Fabric Insert Injection Molding for the Preparation of Ultra High Molecular Weight Polyethylene / High-Density Polyethylene Two-Component Self-reinforced Composites” authored by Jian Wang describes the the ultra-high molecular weight polyethylene (UHMWPE) fabrics were used as the reinforcement insert, and the high-density polyethylene (HDPE) melt was injected to fill the mold cavity and impregnate the fabrics. Furthermore, the results exhibited that the optimum tensile strength and modulus of the UHMWPE/HDPE SrCs were 148 and 1132 MPa, respectively. The peel strength could be up to 35.2 N/cm. The authors report an interesting approach and the presentation of the work is clear. The objective and justification of the work are clear, and the experimental work is significant. The study is accurate and adequate, and thus, I would recommend it for publication in Polymers. However, certain Minor issues are detailed below to improve the quality of the manuscript.

I advise the authors to take the following points into account while revising their manuscript.

Comment 1: There are some typographical errors in the manuscript text, so the authors need to correct them in the revised manuscript. In the whole manuscript, the authors must be taken care of the superscripts and subscripts, and abbreviations. Improve the English language of the manuscript.

Comment 2: The abstract and conclusion sections need to be revised to attain a broad readership.

Comment 3: Minor punctuation revision required in the manuscript.

Comment 4: Please specify the novelty of your study.

Comment 5: The homogeneity of the reference section needs to be maintained. So please check and revise accordingly to the journal's instructions.

Author Response

Thank you very much for the review comments. Accordingly, we have revised the manuscript to address the comments from the reviewers and also improved the writing quality. The detailed point-by-point responses are attached as follows:

Comment 1: There are some typographical errors in the manuscript text, so the authors need to correct them in the revised manuscript. In the whole manuscript, the authors must be taken care of the superscripts and subscripts, and abbreviations. Improve the English language of the manuscript.

Response 1: Thanks for the suggestions. We have modified the manuscript, the superscripts and subscripts have been checked, and some English words and grammar errors have been revised.

Comment 2: The abstract and conclusion sections need to be revised to attain a broad readership.

Response 2: Thanks for the suggestion. We have modified the abstract and conclusions, please see the text in red color.

Comment 3: Minor punctuation revision required in the manuscript.

Response 3: Thanks for the suggestion. We have checked the punctuation in the manuscript and adjusted some sentences and punctuation.

Comment 4: Please specify the novelty of your study.

Response 4: The novelty was described in the abstract and introduction sections. In order to specify the novelty, we have modified and added some sentences. Please see the text in red color.

Comment 5: The homogeneity of the reference section needs to be maintained. So please check and revise accordingly to the journal's instructions.

Response 5: We have checked and revised the references.

Round 2

Reviewer 1 Report

The authors have modified their manuscript as per the comments given by the reviewer.